# Mutations in *NLRP5* and *NLRP9* Are Associated with Litter Size in Small Tail Han Sheep

**DOI:** 10.3390/ani10040689

**Published:** 2020-04-15

**Authors:** Zhuangbiao Zhang, Jishun Tang, Xiaoyun He, Ran Di, Mingxing Chu

**Affiliations:** 1Key Laboratory of Animal Genetics and Breeding and Reproduction of Ministry of Agriculture and Rural Affairs, Institute of Animal Science, Chinese Academy of Agricultural Sciences, Beijing 100193, China; zhangzhuangbiao18@163.com (Z.Z.); tjs157@163.com (J.T.); hedayun@sina.cn (X.H.); diran@caas.cn (R.D.); 2Institute of Animal Husbandry and Veterinary Medicine, Anhui Academy of Agricultural Sciences, Hefei 230031, China

**Keywords:** *NLRP5*, *NLRP9*, SNPs, sheep, litter size, GDF9

## Abstract

**Simple Summary:**

The NLR family pyrin domain-containing 5 (*NLRP5*) and *NLRP9* genes are two important reproductive genes; however, their effects on litter size in sheep are unknown. In this study, we conducted population genetic and association analyses on five *NLRP5* and *NLRP9* loci of sheep. Our results suggested that a mutation in g.60495363G > A may decrease interactions of NLRP5 with proteins, such as the growth differentiation factor 9 (GDF9), whereas a mutation in g. g.59030623T > C may enhance the NLRP9-combining capacity with these proteins. Consequently, these mutations may lead to differences in ovulation rate and even litter size. Overall, this study provided useful genetic markers that can be used to improve sheep breeding.

**Abstract:**

Previous studies showed that the NLR family pyrin domain-containing 5 (NLRP5) and NLRP9 genes are two important reproductive genes; however, their effects on sheep litter size are unknown. Therefore, in this study, we first genotyped seven sheep breeds via the MassARRAY^®^ SNP system at the loci g.60495375A > G, g.60495363G > A, and g.60499690C > A in *NLRP5*, and g.59030623T > C and g.59043397A > C in *NLRP9*. Our results revealed that each locus in most sheep breeds contained three genotypes. Then, we conducted population genetic analysis of single nucleotide polymorphisms in *NLRP5* and *NLRP9*, and we found that the polymorphism information content value in all sheep breeds ranged from 0 to 0.36, and most sheep breeds were under Hardy–Weinberg equilibrium (*p* > 0.05). Furthermore, association analysis in Small Tail Han sheep indicated that two loci, g.60495363G > A in *NLRP5* and g.59030623T > C in *NLRP9*, were highly associated with litter size. The mutation in g.60495363G > A may decrease interactions of NLRP5 with proteins, such as GDF9, whereas the mutation in g.59030623T > C may enhance the combining capacity of NLRP9 with these proteins; consequently, these mutations may influence the ovulation rate and even litter size. The findings of our study provide valuable genetic markers that can be used to improve the breeding of sheep and even other mammals.

## 1. Introduction

Reproduction, a key process in sheep production, is an extremely complex process that is controlled by many genes. NLR family pyrin domain-containing 5 (NLRP5) is an important member of the NLR family and was reported to participate in reproduction in many species. A mutation of *NLRP5* in humans was discovered to be a key factor in maternal reproductive fitness and early zygotic development [1]. Additionally, a lack of *NLRP5* in mouse oocytes resulted in premature activation of the mitochondrial pool, which results in mitochondrial damage that cannot be recovered by BCL2 associated X (*Bax*) inactivation [2], and *NLRP5* knockout in mice led to female infertility [3]. Furthermore, preovulatory aging in mouse oocyte maturation decreased NLRP5 abundance, which indicated that NLRP5 has critical roles in oocyte development [4]. Notably, NLRP5, as a member of the sub-cortical maternal complex, was also found only expressed in ovine ovary and especially in the ovine oocytes in the germinal vesicle and metaphase II stage [5], which suggested their important roles in the oocyte developmental potential of sheep. However, its effects on sheep litter size are poorly understood.

NLRP9 is also an important member of the NLR family. NLRP9 is highly detectable during in vitro maturation of bovine oocytes from small follicles, and may be a factor that affects the follicle diameter [6]. Moreover, *NLRP9* is also highly expressed in the ovine ovary [7]; however, the effects of *NLRP9* on sheep reproduction, such as litter size, remain largely unknown.

Significantly, point mutations of several genes affecting the litter size in sheep were discovered. Several were found to be major genes in reproduction; the most prominent gene, *FecB* (Fec = Fecundity, B = Booroola), was identified as a missense mutation in bone morphogenetic protein receptor, type 1B (*BMPR1B*) at base A746G, and this mutation results in an amino acid change from glutamine to arginine [8]. The effects of this mutation on ovulation and litter size mainly depend on the copy carried. Normally, ewes with one copy of the *FecB* mutation increase the rate of ovulation by 1.5 and litter size by 1, and ewes with two copies can significantly increase the rate of ovulation and litter size by 3 and 1.5 (summarized by Liu et al. [9]). In addition, several other mutations in genes, including the growth differentiation factor 9 gene (*GDF9*) [10,11], bone morphogenetic protein 15 (*BMP15*) [12], *BMP2*, and *BMP7* [13], were also reported to participate in sheep reproduction and lead to differences in litter size; all of these point mutations could be important genetic markers for sheep breeding.

Therefore, in this study, we aim to explore the association between *NLRP5* and *NLRP9* single nucleotide polymorphisms (SNPs) with litter size. This information has not previously been elucidated but is expected to provide valuable genetic markers for sheep breeding.

## 2. Materials and Methods

### 2.1. Animal Preparation and Sample Collection

All experimental procedures involving animals used in this study were approved by the Science Research Department (in charge of animal welfare issues) of the Institute of Animal Science, Chinese Academy of Agricultural Sciences (IAS-CAAS; Beijing, China). In addition, ethics approval was given by the animal ethics committee of IAS-CAAS (no. IASCAAS-AE-03, 12 December 2016).

Blood samples from 768 ewes were obtained from the jugular vein, and all samples were processed with the phenol–chloroform method for DNA isolation. Among the seven sheep breeds used in this study, three are higher prolificacy breeds (Cele Black sheep, *n* = 68; Hu sheep, *n* = 83; and Small Tail Han sheep, *n* = 384) and four breeds are lower prolificacy breeds (Prairie Tibetan sheep, *n* = 80; Suffolk sheep, *n* = 60; Sunite sheep, *n* = 70; and Tan sheep, *n* = 23) (Table 1).

### 2.2. Genotyping

First, single-base extended primers for g.60495375A > G, g.60495363G >A, and g.60499690C > A in *NLRP5* and g.59030623T > C and g.59043397A > C in *NLRP9* were designed via MassARRAY Assay Design v. 3.1 based on the sheep sequences of *NLRP5* and *NLRP9* available in GenBank (accession no.: NC_019471.1, *NLRP5*; NC_019471.1, *NLRP9*). Then, these primers were synthesized by Beijing Compass Biotechnology Co., Ltd. (Beijing, China). The genotyping was conducted in a MassARRAY^®^ SNP system; detailed information about the system and procedures has been previously described [6,14].

### 2.3. Statistical Analysis

Allele and genotype frequency, polymorphism information content (PIC), heterozygosity (HE), and number of effective alleles (NE), and *p* values (Chi-Square test) were calculated using the data after genotyping, and ewe populations with *p* > 0.05 (Chi-Square test) were considered to be under Hardy–Weinberg equilibrium. To determine the association between genotypes and litter size, the adjusted linear model, *y_ijn_* = *μ*+ *P_i_*+*G_j_* + *I_PG_* + *e_ijn_*, was applied, in which *y_ijn_* is the phenotypic value (litter size); *μ* represents the population mean; *P_i_* indicates the fixed effect of the *i^th^* parity (*i* = 1, 2, or 3); *G_j_* represents the effect of the *j^th^* genotypes (*j* = 1, 2, or 3); *I_PG_* is the interactive effect of parity and genotype; and *e_ijn_* indicates random error.

### 2.4. Protein Interaction Networks Predicted by STRING Database

To primarily explore the mechanisms by which NLRP5 and NLRP9 affect litter size, we predicted the protein interactions involving NLRP5 and NLRP9 via the STRING database v. 11.0 [15] (https://string-db.org), which is a useful tool to predict protein interactions. We firstly typed the protein name, then selected the organism sheep (*Ovis aries*), then carried out the prediction according to the default settings (meaning of network edges: evidence).

## 3. Results

### 3.1. Population Genetic Analysis of SNPs in NLRP5 and NLRP9

Three SNPs in *NLPR5* (g.60495375A > G, g.60495363G>A, and g.60499690C > A) and two SNPs in *NLRP9* (g.59030623T > C and g.59043397A > C) were detected (Table 2 and Table 3). The g.60495375A > G locus had low polymorphism (PIC < 0.25) in Tan, Small Tail Han, and Suffolk sheep but had moderate polymorphism (0.25 < PIC < 0.5) in the Prairie Tibetan, Cele Black, Hu, and Sunite sheep; in addition, the Chi-square test revealed that this locus was under Hardy–Weinberg equilibrium in Cele Black, Hu, Sunite, Small Tail Han, and Suffolk sheep (*p* > 0.05) but not in Prairie Tibetan and Tan sheep (*p* < 0.05). The g.60495363G > A locus was moderately polymorphic (0.25 < PIC < 0.5) in all seven sheep breeds, and this SNP was under Hardy–Weinberg equilibrium in all seven sheep breeds (*p* > 0.05). In all seven sheep breeds, the g.60499690C > A locus was moderately polymorphic (0.25 < PIC < 0.5) and under Hardy–Weinberg equilibrium (*p* > 0.05). The g.59030623T > C locus had relatively low polymorphism (PIC < 0.25) in Prairie Tibetan, Cele Black, Hu, Small Tail Han, and Tan sheep but had moderate polymorphism (0.25 < PIC < 0.5) in Suffolk and Sunite sheep; additionally, this SNP was under Hardy–Weinberg equilibrium in all seven sheep breeds (*p* > 0.05). The g.59043397A > C locus had relatively low polymorphism (PIC < 0.25) in Prairie Tibetan, Suffolk, Sunite, Small Tail Han, and Tan sheep but had moderate polymorphism (0.25 < PIC < 0.5) in Cele Black and Hu sheep; furthermore, this locus was under Hardy–Weinberg equilibrium (*p* > 0.05) in all sheep breeds except Small Tail Han sheep (*p* < 0.05).

### 3.2. Associations between Five Loci in NLRP5 and NLRP9 with Litter Size in Small Tail Han Sheep

The results (Table 4) revealed that the g.60495363G > A locus in *NLRP5* was highly associated with litter size in Small Tail Han sheep; the litter size of ewes with the GG genotype was higher than that of ewes with AA and GA genotypes. Additionally, the g.59030623T > C locus in *NLRP9* was highly associated with litter size in Small Tail Han sheep; the litter size of ewes with the CC genotype was higher than that of ewes with TT and TC genotypes (Table 5). The remaining three loci had no significant association with litter size in Small Tail Han sheep.

### 3.3. Predicted Protein Interaction Networks Involving NLRP5 and NLRP9

As Figure 1 shows, NLRP5 was predicted to interact with 10 proteins, including GDF9, which has been proven to be a key factor that affects litter size [16,17]. Interestingly, NLRP9 was also predicted to interact with 10 proteins, including GDF9.

## 4. Discussion

### 4.1. NLRP5 and NLRP9 Polymorphisms

As two important members of the NLR family, NLRP5 and NLRP9 play important roles in mammal reproduction [1,11,12]. However, little is known about their effects on litter size in ewes. Therefore, in this study, we first conducted population genetic analysis of five loci in *NLRP5* and *NLRP9*. The results suggested that the loci in some sheep breeds exhibited relatively low polymorphism (PIC < 0.25), such as g.60495375A > G, g.59030623T > C, and g.59043397A > C in Tan sheep, and g.59030623T > C and g.59043397A > C in Prairie Tibetan sheep; this may result from the examination of a limited number of ewes, and increasing the number of examined ewes may increase the PIC value. Additionally, many ewe populations contained three genotypes, which indicated that those SNPs were widely distributed in sheep herds that include sheep breeds with different fecundity. However, only two genotypes were detected in some cases, such as for g.59030623T > C in Tan and Hu sheep, and three genotypes may be detected by increasing the number of examined ewes. In addition, several loci, such as g.60495375A > G in Tan sheep and g.59043397A > C in Small Tail Han sheep, were not in Hardy–Weinberg equilibrium (*p* < 0.05), which may result from natural and artificial selection.

### 4.2. Association Analyses of NLRP5 and NLRP9

Early studies grouped *NLRP5* and *NLRP9* as two members of the reproduction-related *NLRP* cluster in mammals [18], which indicates that they have potential functions in reproduction. Docherty et al. [1] reported that some *NLRP5* variants in women were associated with a period of infertility, miscarriage, and molar pregnancy, which may be consequences of reproductive problems. In our results, three SNPs were detected; two SNPs showed an increasing litter size trend but failed to reach significance (*p* > 0.05), whereas the wildtype homozygous genotype of g.60495363G > A (GG) was associated with significantly greater litter size in Small Tail Han sheep compared with GA and AA genotypes; therefore, this mutation seems to be harmful.

Previous studies found that several variants in *NLRP9* may influence the disease course in multiple sclerosis patients, as determined by an exome sequencing study [19], and may be associated with familial late-onset Alzheimer’s disease, as determined by a genome-wide association study [20]. However, little has been found regarding its effects on litter size. In our study, two detected SNPs showed increasing trends in litter size; in particular, a mutation in the g.59030623T > C locus could highly increase litter size in Small Tail Han sheep, which may be a useful genetic marker for increasing sheep litter size.

### 4.3. Reproductive Functions of GDF9 and Its Interactions with NLRP5 and NLRP9

GDF9 has been proven to be highly associated with litter size in sheep (Small Tail Han sheep) [10,21], goats [22,23], and even dogs [24]. Ongoing research has revealed some potential mechanisms underlying this association. GDF9 was reported to enhance mitochondrial activity, meiotic resumption, and secondary follicle development in sheep [25], and GDF9 was also shown to enhance granulosa cell proliferation [26]. Further study indicated that GDF9 is required for normal folliculogenesis in many mammal species [17]; therefore, GDF9 may be a direct factor that influences litter size in ewes by affecting functions of granulosa cells and folliculogenesis.

NLRP5 in ovine species was demonstrated to cooperate with oocyte expressed protein (OOEP), TLE family member 6 (TLE6), and KH domain containing 3 (KHDC3) to function in oocyte development [5], and it was also proven to meditate the mitochondrial function in mice, which is indispensable for oocyte development [2]. Moreover, NLRP5, also called MATER, could mediate follicular maturation in humans by acting as a substrate of protein kinase C epsilon in cumulus cells [27]. It is possible that a missense mutation from G to A at the g.60495363G > A locus may decrease its positive effects on oocyte development and follicular maturation. Little is known about the effects of NLRP9 on oocyte development or follicular maturation; however, researches demonstrated that *NLRP9* was highly expressed in adult bovine oocytes rather than prepubertal animals [12] and was particularly highly expressed in sheep ovaries [7]. Additionally, two oocyte-specific NLRP genes in mammals, *NLRP5* and *NLRP9,* were only expressed in mouse oocytes [28], which indicated that they have critical roles in ovary functions, such as oocyte development and follicular maturation. Therefore, by considering the key roles of NLRP5 and NLRP9 in ovine ovary functions, such as oocyte development and follicular maturation, predicted protein interactions, and point mutation effects on the protein-binding capacity [29,30], we conclude that the missense mutation in g.60495363G > A of *NLRP5* may decrease the interactions of NLRP5, such as the combining capacity, with proteins, such as GDF9; this may decrease the litter size. Conversely, the missense mutation in g.59030623T > C of *NLRP9* may enhance the NLRP9 combining capacity with proteins, such as GDF9, which would promote the ovulation rate and even increase the litter size.

## 5. Conclusions

In this study, we first conducted population genetic analysis of SNPs in *NLRP5* and *NLRP9*. We found two key loci (g.60495363G > A in *NLRP5* and g.59030623T > C in *NLRP9*) via association analysis. Further analysis indicated that these detected missense mutations may influence litter size in ewes by affecting the interactions of NLRP5 and NLRP9 with proteins, such as GDF9.

## Figures and Tables

**Figure 1 animals-10-00689-f001:**
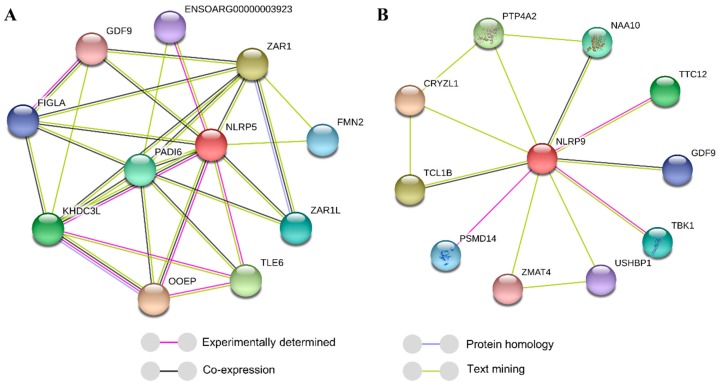
Interactions between proteins including NLRP5 (**A**) and NLRP9 (**B**), as predicted by the STRING database.

**Table 1 animals-10-00689-t001:** Basic information on ewes used in this study.

Breed	Number	Type	District
Small Tail Han sheep	384	Year-round Breeding	Southwest region, Shandong Province, China
Hu sheep	83	Year-round Breeding	Xuzhou, Jiangsu Province, China
Cele Black sheep	68	Year-round Breeding	Cele, Xinjiang Uygur Autonomous Region, China
Sunite sheep	70	Seasonal Breeding	Wulatezhongqi, Bayannaoer, Inner Mongolia Autonomous Region, China
Prairie Tibetan sheep	80	Seasonal Breeding	Dangxiong, Tibet Autonomous Region, China
Suffolk sheep	60	Seasonal Breeding	Beijing Aoxin Stud Farm Co. Ltd. located in Shunyi District, Beijing, China
Tan sheep	23	Seasonal Breeding	Yanchi, Ningxia Hui Autonomous Region, China

**Table 2 animals-10-00689-t002:** Population genetic analysis of three NLR family pyrin domain-containing 5 (*NLRP5*) loci in seven sheep breeds.

Locus	Breed	Genotype Frequency			Allele Frequency		PIC	HE	NE	*Chi*-Square Test (*p*-Value)
g.60495375 A > G		**AA**	**AG**	**GG**	**A**	**G**				
	Prairie Tibetan sheep	0.61	0.25	0.14	0.74	0.26	0.31	0.39	1.63	0.00
	Cele Black sheep	0.62	0.34	0.04	0.79	0.21	0.28	0.34	1.50	0.95
	Hu sheep	0.52	0.35	0.13	0.69	0.31	0.34	0.43	1.74	0.10
	Suffolk sheep	0.75	0.20	0.05	0.85	0.15	0.23	0.26	1.35	0.10
	Sunite sheep	0.54	0.37	0.09	0.73	0.27	0.32	0.40	1.65	0.61
	Tan sheep	0.87	0.09	0.04	0.91	0.09	0.15	0.16	1.19	0.03
	Small Tail Han Sheep	0.74	0.24	0.02	0.86	0.14	0.21	0.24	1.32	0.87
g.60495363 G > A		**AA**	**AG**	**GG**	**A**	**G**				
	Prairie Tibetan sheep	0.62	0.35	0.03	0.80	0.20	0.27	0.32	1.47	0.40
	Cele Black sheep	0.46	0.46	0.08	0.68	0.32	0.34	0.43	1.76	0.65
	Hu sheep	0.47	0.42	0.11	0.68	0.32	0.34	0.43	1.77	0.79
	Suffolk sheep	0.26	0.52	0.22	0.52	0.48	0.37	0.50	2.00	0.78
	Sunite sheep	0.60	0.36	0.04	0.78	0.22	0.29	0.34	1.53	0.76
	Tan sheep	0.65	0.26	0.09	0.78	0.22	0.28	0.34	1.52	0.26
	Small Tail Han Sheep	0.45	0.43	0.12	0.66	0.34	0.35	0.45	1.81	0.52
g.60499690 C > A		**CC**	**CA**	**AA**	**C**	**A**				
	Prairie Tibetan sheep	0.39	0.46	0.15	0.62	0.38	0.36	0.47	1.89	0.86
	Cele Black sheep	0.29	0.56	0.15	0.57	0.43	0.37	0.49	1.96	0.24
	Hu sheep	0.22	0.46	0.32	0.45	0.55	0.37	0.49	1.98	0.50
	Suffolk	0.53	0.42	0.05	0.73	0.27	0.31	0.39	1.64	0.40
	Sunite sheep	0.40	0.44	0.16	0.62	0.38	0.36	0.47	1.89	0.62
	Tan sheep	0.57	0.39	0.04	0.76	0.24	0.30	0.36	1.57	0.72
	Small Tail Han Sheep	0.47	0.41	0.12	0.67	0.33	0.34	0.44	1.78	0.22

**Table 3 animals-10-00689-t003:** Population genetic analysis of two loci of *NLRP9* in seven sheep breeds.

Locus	Breed	Genotype Frequency			Allele Frequency		PIC	HE	NE	*Chi*-Square Test (*p*-Value)
g.59030623 T > C		**TT**	**TC**	**CC**	**T**	**C**				
	Prairie Tibetan sheep	0.81	0.18	0.01	0.90	0.10	0.16	0.18	1.22	0.80
	Cele Black sheep	0.91	0.09	0.00	0.96	0.04	0.08	0.08	1.09	0.70
	Hu sheep	0.75	0.25	0.00	0.87	0.13	0.20	0.22	1.28	0.19
	Suffolk	0.52	0.38	0.10	0.71	0.29	0.33	0.41	1.70	0.58
	Sunite sheep	0.63	0.37	0.00	0.81	0.19	0.26	0.30	1.43	0.06
	Tan sheep	0.83	0.17	0.00	0.91	0.09	0.15	0.16	1.19	0.65
	Small Tail Han Sheep	0.74	0.23	0.03	0.85	0.15	0.22	0.25	1.34	0.07
g.59043397 A > C		**AA**	**CA**	**CC**	**A**	**C**				
	Prairie Tibetan sheep	0.86	0.13	0.01	0.92	0.08	0.13	0.14	1.16	0.38
	Cele Black sheep	0.56	0.37	0.07	0.74	0.26	0.31	0.38	1.62	0.75
	Hu sheep	0.48	0.36	0.16	0.66	0.34	0.35	0.45	1.81	0.08
	Suffolk	0.95	0.05	0.00	0.98	0.02	0.05	0.05	1.05	0.84
	Sunite sheep	0.83	0.16	0.01	0.91	0.09	0.15	0.17	1.20	0.57
	Tan sheep	0.65	0.35	0.00	0.83	0.17	0.25	0.29	1.40	0.31
	Small Tail Han Sheep	0.78	0.20	0.02	0.88	0.12	0.19	0.22	1.28	0.05

**Table 4 animals-10-00689-t004:** Least squares mean and standard error of litter size in Small Tail Han sheep with different genotypes of g.60495375A > G, g.60495363G > A, and g.60499690C > A.

Locus	Genotype	Litter Size		
		First Parity (N)	Second Parity (N)	Third Parity (N)
g.60495375 A > G	AA	1.88 ± 0.039(283)	2.12 ± 0.058(170)	2.40 ± 0.101(68)
	AG	1.83 ± 0.069(92)	2.11 ± 0.103(54)	2.53 ± 0.191(19)
	GG	2.00 ± 0.234(8)	2.40 ± 0.340(5)	3.00 ± 0.416(4)
g.60495363 G > A	AA	1.81 ± 0.051(169) ^b^	2.06 ± 0.077(97) ^b^	2.37 ± 0.133(38) ^b^
	GA	1.89 ± 0.052(164) ^a,b^	2.07 ± 0.078(94) ^b^	2.33 ± 0.137(36) ^b^
	GG	2.12 ± 0.098(46) ^a^	2.37 ± 0.128(35) ^a^	2.86 ± 0.205(16) ^a^
g.60499690 C > A	CC	1.86 ± 0.049(180)	2.10 ± 0.073(107)	2.49 ± 0.137(37)
	CA	1.85 ± 0.053(158)	2.07 ± 0.077(98)	2.36 ± 0.137(42)
	AA	2.00 ± 0.098(46)	2.10 ± 0.073(24)	2.67 ± 0.241(12)

Different letters indicate significant difference (*p* < 0.05).

**Table 5 animals-10-00689-t005:** Least squares mean and standard error of litter size in Small Tail Han sheep with different genotypes of g.59030623T > C and g.59043397A > C.

Locus	Genotype	Litter Size		
		First Parity (N)	Second Parity (N)	Third Parity (N)
g.59030623 T > C	TT	1.85 ± 0.039(283) ^b^	2.10 ± 0.062(152) ^b^	2.34 ± 0.120(47) ^b^
	TC	1.88 ± 0.071(88) ^b^	2.13 ± 0.095(64) ^b^	2.46 ± 0.139(35) ^b^
	CC	2.25 ± 0.200(11) ^a^	2.55 ± 0.229(11) ^a^	3.00 ± 0.273(9) ^a^
g.59043397 A > C	AA	1.87 ± 0.038(298)	2.10 ± 0.056(186)	2.47 ± 0.097(75)
	AC	1.85 ± 0.077(75)	2.18 ± 0.124(38)	2.27 ± 0.223(14)
	CC	2.20 ± 0.210(10)	2.25 ± 0.381(4)	3.00 ± 0.591(2)

Different letters indicate significant difference (*p* < 0.05).

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
