# Peer review of "Mutations in NLRP5 and NLRP9 Are Associated with Litter Size in Small Tail Han Sheep"

_animals, 2020, doi:10.3390/ani10040689_

Round 1

Reviewer 1 Report

In general:
The article evaluates the population genetic parameters for 5 single nucleotide variants (SNVs) in the genes NLRP5 (3 SNVs) and NLRP9 (2 SNVs) across 7 sheep breeds, in order to identify their influence on litter size. The study was conceived robustly based on the lack of information concerning these two genes in sheep fecundity. The study was design with enough power to back the conclusions and the conclusions describe the observations well enough.
One idea that is not clear however is how the 5 markers were chosen.
The article would benefit greatly with a thorough revision of the writing, ideally by a native speaker. I tried to point out the mistakes at the beginning of reading the article, but they were too abundant.
I would recommend the publication of this article in MDPI-Animals after minor revisions.

Specifics:
line16: In simple summary, the sentence "results suggested that a mutation in 16 g.60495363G>A may decrease its interaction with proteins like GDF9, while a mutation in 17 g.59043397A>C may enhance its combining capacity with proteins like GDF9" is ambiguous. Does "its" refer to NLRP5 or NLRP9?

line29: The sentence "Furthermore, association analysis in Small Tail Han sheep indicated two loci of g.60495363G>A in NLRP5 and g.59043397A>C in NLRP9 were highly associated with litter size" is poorly written. Please change accordingly to indicate that two loci of interest were found: g.60495363G>A in NLRP5 and g.59043397A>C in NLRP9.

line31: Poorly written sentence: "mutation occurred in g.60495363G>A may decrease its interaction with proteins like GDF9, while 31 a mutation occurred in g.59043397A>C may". Please use "occurring"

line93: it seems that the "Pj" should be "Pi" as in the formula.

line96: the section 2.4 would improve by including a description of the STRING data base and the parameters under which exploration was conducted.

Author Response

Responds to the reviewer’s comments

In general:

  1. The article evaluates the population genetic parameters for 5 single nucleotide variants (SNVs) in the genes NLRP5 (3 SNVs) and NLRP9 (2 SNVs) across 7 sheep breeds, in order to identify their influence on litter size. The study was conceived robustly based on the lack of information concerning these two genes in sheep fecundity. The study was design with enough power to back the conclusions and the conclusions describe the observations well enough.

One idea that is not clear however is how the 5 markers were chosen.

The article would benefit greatly with a thorough revision of the writing, ideally by a native speaker. I tried to point out the mistakes at the beginning of reading the article, but they were too abundant.

I would recommend the publication of this article in MDPI-Animals after minor revisions.

Response: We thank you very much for your positive comments on our manuscript. The 5 significantly markers in NLRP5 and NLRP9 were detected in Small Tail Han Sheep according to our early study on variation of the genome in 89 sheep by re-sequencing, and the several others significantly SNPs have published in our early reports (Pan et al., 2018; Zhou et al., 2018, La et al., 2019). Additionally, the manuscript has been revised by native speaker thoroughly, if you need it, I can provide the license.

[1] Pan Z, Li S, Liu Q et al. Whole-genome sequences of 89 Chinese sheep suggest role of RXFP2 in the development of unique horn phenotype as response to semi-feralization. Gigascience, 7(4):1-15 .DOI:10.1093/gigascience/giy019.

[2] Zhou Mei, Pan Zhangyuan, Cao Xiaohan et al. Single nucleotide polymorphisms in the HIRA gene affect litter size in Small Tail Han sheep, Animals, 2018, 8(5), 71; DOI: 10.3390/ani8050071.

[3] La Yongfu, Liu Qiuyue, Zhang Liping et al. Single nucleotide polymorphisms in SLC5A1, CCNA1, and ABCC1 and the association with litter size in Small Tail Han sheep. Animals, 2019, 9(7), 432. DOI: 10.3390/ani9070432.

Specifics:

  1. line16: In simple summary, the sentence "results suggested that a mutation in 16 g.60495363G>A may decrease its interaction with proteins like GDF9, while a mutation in 17 g.59043397A>C may enhance its combining capacity with proteins like GDF9" is ambiguous. Does "its" refer to NLRP5 or NLRP9?

Response: We thank you very much for your carefulness, the sentence has been revised as “Our results suggested that a mutation in g.60495363G>A may decrease interactions of NLRP5 with proteins such as GDF9, whereas a mutation in g.59043397A>C may enhance NLRP9 combining capacity with these proteins”.

  1. line29: The sentence "Furthermore, association analysis in Small Tail Han sheep indicated two loci of g.60495363G>A in NLRP5 and g.59043397A>C in NLRP9 were highly associated with litter size" is poorly written. Please change accordingly to indicate that two loci of interest were found: g.60495363G>A in NLRP5 and g.59043397A>C in NLRP9.

Response: We thank you very much for your carefulness, this sentence has been revised as “association analysis in Small Tail Han sheep indicated that two loci, g.60495363G>A in NLRP5 and g.59043397A>C in NLRP9, were highly associated with litter size ".

  1. line31: Poorly written sentence: "mutation occurred in g.60495363G>A may decrease its interaction with proteins like GDF9, while 31 a mutation occurred in g.59043397A>C may". Please use "occurring"

Response: We thank you very much for your carefulness, it has been revised accordingly.

  1. line93: it seems that the "Pj" should be "Pi" as in the formula.

Response: We thank you very much for your carefulness, the "Pj" has been revised as "Pi" in the formula.

  1. line96: the section 2.4 would improve by including a description of the STRING data base and the parameters under which exploration was conducted.

Response: We thank you very much for your carefulness, this section has bee revised as “To primarily explore the mechanisms by which NLRP5 and NLRP9 affect litter size, we predicted the protein interactions involving NLRP5 and NLRP9 via STRING database v. 11.0 [15] (https://string-db.org), which is a useful tool to predict protein interactions. We firstly typed the protein name, then selected the organism-sheep (Ovis aries), then carried out the prediction according to the default settings (meaning of network edges: evidence).

Reviewer 2 Report

This paper performs an association study of SNPs in two genes previously associated with reproductive functions. This is a short manuscript and for the most part is clear to read. My concerns relate primarily to presentation although I would also comment on the methods: to create more robust associations, it would be worth running the regression analysis only using genotypes from those animals which give birth to live young (i.e. that provide the litter size data), the latter being a subset of the former (it is already suggested on line 154 that there are a limited number of ewes in certain cases, with a few very low sample sizes – and accordingly insignificant associations – in table 5).

The start of the introduction also places emphasis on genes not actually included within the study (lines 38-47) and so it is not immediately clear why NLRP5 and NLRP9 are being focused upon here, nor why a functional effect on litter size (or, later in the paper, ovulation rate) would specifically be expected – the referenced material is not overly detailed (for instance, line 56 states that NLRP5 “plays important roles in the oocyte developmental potential” but there is little elaboration). Similarly, is there some expectation that these SNPs will introduce a meaningful structural change to their proteins – are they all actually non-synonymous polymorphisms? It would benefit the introduction/discussion to include some mention of this. For instance, the NRLP5 mutation, 60495363G>A, is missense (rs417878239), consistent with the conclusion on line 200.

Line 100. This lengthy URL can be reduced to https://string-db.org

Line 104. These references to figures 2 and 3 come before the first reference to figure 1 (on line 139).

Tables 4 and 5. It isn’t clear to me why significant differences are labelled ‘a’, ‘b’ and ‘ab’ as what these letters refer to isn’t stated. My initial assumption was that they referred to pairwise comparisons across different rows in each column, i.e. that there was a significant difference between the two entries labelled a, and so on. However, this cannot be the case as not every row contains equal numbers of ‘a’ and ‘b’. Can you please elaborate?

In addition, it would be clearer to add the number in parentheses after the mean +/- SE; “mean (n) +/- SE” is somewhat atypical.

Line 144. What do the colours of each node refer to in figure 1? The legend shows only edge colours.

Finally, while for the most part clear to read, there are numerous grammatical errors, some of which are noted below, although this is a non-exhaustive list. While this does not greatly affect a technical reading of the paper, additional proofreading would be beneficial prior to publication.

Line 27. "was ranged" should be "ranged"

Line 42. "were mainly depend" should be "depend"

Line 43. "increase ovulation by 1.5" - I assume "increase the rate of ovulation by 1.5x"?

Lines 48 and 58. "member of NLR family" should be "member of the NLR family"

Line 104. "Totally" should be "In total".

Line 149. “according previous studies” should be “according to previous studies”

Author Response

Responds to the reviewer’s comments

  1. This paper performs an association study of SNPs in two genes previously associated with reproductive functions. This is a short manuscript and for the most part is clear to read. My concerns relate primarily to presentation although I would also comment on the methods: to create more robust associations, it would be worth running the regression analysis only using genotypes from those animals which give birth to live young (i.e. that provide the litter size data), the latter being a subset of the former (it is already suggested on line 154 that there are a limited number of ewes in certain cases, with a few very low sample sizes – and accordingly insignificant associations – in table 5).

Response: We thank you very much for your positive comments on our manuscript. Generally, the number of sheep used in this study is relatively large sheep population, though some sheep breeds like Tan sheep was a limited number, luckily, all the genotypes has been detected, which revealed that these mutations were actually existed, suggesting the value of this study for animal breeding. In addition, compared with other researches similar to our study and published in Animals (Zhou et al., (2018), La et al., (2019)), the number of ewes used in this study was rational. In the future study, we will still focus on these SNPs, and try to understand the potential mechanism by which these SNPs affect litter size.

  1. Zhou Mei, Pan Zhangyuan, Cao Xiaohan et al. Single Nucleotide Polymorphisms in the HIRA Gene Affect Litter Size in Small Tail Han Sheep, Animals 2018, 8(5), 71; https://doi.org/10.3390/ani8050071.
  2. La Yongfu, Liu Qiuyue, Zhang Liping et al. Single Nucleotide Polymorphisms in SLC5A1, CCNA1, and ABCC1 and the Association with Litter Size in Small Tail Han Sheep. Animals 2019, 9(7), 432; https://doi.org/10.3390/ani9070432.

  1. The start of the introduction also places emphasis on genes not actually included within the study (lines 38-47) and so it is not immediately clear why NLRP5 and NLRP9 are being focused upon here, nor why a functional effect on litter size (or, later in the paper, ovulation rate) would specifically be expected – the referenced material is not overly detailed (for instance, line 56 states that NLRP5 “plays important roles in the oocyte developmental potential” but there is little elaboration). Similarly, is there some expectation that these SNPs will introduce a meaningful structural change to their proteins – are they all actually non-synonymous polymorphisms? It would benefit the introduction/discussion to include some mention of this. For instance, the NRLP5 mutation, 60495363G>A, is missense (rs417878239), consistent with the conclusion on line 200.

Response: We thank you very much for your carefulness, the introduction has been revised and some words of “non-synonymous mutation” has been added in discussion part in Line 196-198 accordingly. In addition, the referenced material has been detailly described in further, details see the revised manuscript.

  1. Line 100. This lengthy URL can be reduced to https://string-db.org

Response: We thank you very much for your carefulness, it has been revised accordingly.

  1. Line 104. These references to figures 2 and 3 come before the first reference to figure 1 (on line 139).

Response: We thank you very much for your carefulness, it has been revised accordingly.

  1. Tables 4 and 5. It isn’t clear to me why significant differences are labelled ‘a’, ‘b’ and ‘ab’ as what these letters refer to isn’t stated. My initial assumption was that they referred to pairwise comparisons across different rows in each column, i.e. that there was a significant difference between the two entries labelled a, and so on. However, this cannot be the case as not every row contains equal numbers of ‘a’ and ‘b’. Can you please elaborate?

Response: We thank you very much for your carefulness, the comparisons was conducted in the same locus and same parity but different genotypes. For instance, in g.60495363G>A, the litter size in sheep with AA in the first parity was labeled as ‘b’, while the litter size in sheep with GG in the first parity was labeled as ‘a’, which indicates that there is significant difference (no same letter) between AA and GG genotypes in litter size (P<0.05). Similarly, there is no significant difference (having same letter) between AA (‘a’) and GA (‘ab’) genotypes in litter size (P>0.05).

  1. In addition, it would be clearer to add the number in parentheses after the mean +/- SE; “mean (n) +/- SE” is somewhat atypical.

Response: We thank you very much for your carefulness, it has been revised accordingly.

  1. Line 144. What do the colours of each node refer to in figure 1? The legend shows only edge colours.

Response: We thank you very much for your carefulness, the different colors of each node refer to different proteins in same network, the lines between two proteins represents how the interactive relationships was built.

  1. Finally, while for the most part clear to read, there are numerous grammatical errors, some of which are noted below, although this is a non-exhaustive list. While this does not greatly affect a technical reading of the paper, additional proofreading would be beneficial prior to publication.

Response: We thank you very much for your carefulness, the manuscript has been revised by native speaker thoroughly, if you need it, I can provide the license.

  1. Line 27. "was ranged" should be "ranged"

Response: We thank you very much for your carefulness, it has been revised accordingly.

  1. Line 42. "were mainly depend" should be "depend"

Response: We thank you very much for your carefulness, it has been revised accordingly.

  1. Line 43. "increase ovulation by 1.5" - I assume "increase the rate of ovulation by 1.5x"?

Response: We thank you very much for your carefulness, it has been revised accordingly.

  1. Lines 48 and 58. "member of NLR family" should be "member of the NLR family"

Response: We thank you very much for your carefulness, it has been revised accordingly.

  1. Line 104. "Totally" should be "In total".

Response: We thank you very much for your carefulness, it has been revised accordingly.

  1. Line 149. “according previous studies” should be “according to previous studies”

Response: We thank you very much for your carefulness, it has been revised accordingly.